# "Good Foreign Language Teachers Pay Attention to Heterogeneity": Conceptualizations of Differentiation and Effective Teaching Practice in Inclusive EFL Classrooms by German Pre-Service Teachers

Ana Rovai and Joanna Pfingsthorn *

Department of English Language Education, University of Bremen, 28359 Bremen, Germany;
ana.rovai@uni-bremen.de
* Correspondence: pfingsthorn@uni-bremen.de

**Abstract:** This paper explores how pre-service EFL teachers perceive the variety of methodic-didactic and pedagogical forms of differentiation that they consider as acceptable in their teaching practice and which shed light on knowledge areas related to adaptivity competence. Our investigation looks into (a) qualitative questionnaire data that depict pre-service FL teachers' conceptualizations of what it means to be a "good" and "bad" foreign language teacher; and (b) pre-service FL teachers' quantitative evaluations of existing differentiation approaches designed for accommodating learners, especially ones experiencing specific learning differences such as difficulties with memorization, classroom communication, anxiety, or lexical and grammar confusion. Our results show that, despite expressing general agreement towards supporting individual learners' needs, participants' knowledge regarding how to respond to the needs of all FL learners appropriately is incomplete.

**Keywords:** foreign language education; inclusive education; teachers' adaptivity competence; learner diversity

## 1. Introduction

The progressive implementation of inclusive education in Germany has heightened the need for foreign language (FL) education to find effective ways to embrace learners' diversity and avoid the contextual reproduction of learning barriers (Rossa 2015). The goal is to make learning and participation accessible to all (Booth and Ainscow 2003, p. 10). However, as teaching English as a FL to students with special educational needs was long regarded as not worth pursuing in Germany (cf. Dose 2019; Kleinert et al. 2007; Morse 2008) and because of the relatively long tradition of general segregationist schooling practice (Pfahl and Powell 2011), inclusive FL education in Germany has not yet completed the task of questioning previous beliefs about diversity and effective foreign language teaching and learning (Rossa 2015).

Given that teachers' actions can significantly influence the learning progress of students (Hattie 2009), their adaptivity towards heterogeneity is crucial to establishing inclusive learning environments. The concept of "adaptivity" is understood here as teachers' flexible use of various teaching approaches that offer appropriate support for individual learning processes (Gerlach and Leupold 2019, p. 93). It implies that FL teachers are in the position to take a step back from pre-established ideas of a "good method" or a "good teacher" and situationally respond to the needs of learners (Gerlach and Leupold 2019, p. 25).

Adaptivity competence is especially relevant for teaching languages to learners with specific learning differences (SpLDs) who may experience difficulties in phonological processing, word recognition, metacognitive language learning awareness, lack of attention

or anxiety, and demotivation (Heimlich 2016; Nijakowska 2010) or have a negative self-concept and negative attitudes towards the target language (Csizer et al. 2010; Nijakowska 2010). This implies that teachers may be required to identify moments in which it is necessary to steer away from typical, open communicative tasks and to reflect on the individualization of measures and gradation of task complexity (Nijakowska 2010). They can, instead, choose, for instance, a sequential, systematic, and cumulative multisensory approach that includes systematic and recurrent drills targeting phonological awareness, the metacognitive component of vocabulary, or tasks that focus on explicit textual structure (Daloiso 2017; Kormos and Smith 2012; Nijakowska 2010).

Thus, in this study, we investigate the extent to which pre-service FL teachers, who are pursuing a graduate teaching degree for inclusive schools, acknowledge the potential positive effect of differentiated instruction on learning, even if some approaches may seem counter-intuitive to what traditional communicative FL teaching implies. In addition, we examine pre-service FL teachers' general conceptualization of what it means to be a "good" and a "bad teacher", which provides insights into teacher cognition. A consensus is that teachers' "mental lives", i.e., what they know and think, as well as their attitude and belief system, exert a substantial influence on their pedagogical practice and decision-making processes (Borg 2003, 2011). In line with Borg (2003, 2011), this study, thus, taps into aspects of teacher cognition through self-reports to shed light on how future English FL teachers, expected to teach inclusively, conceptualize "good" and "bad" teaching practice, as well as capture their perceptions of adaptivity competence in terms of differentiation approaches.

### 1.1. Inclusion and Specific Learning Differences

Inclusion, in contrast to integration, of people with special needs into regular schools assumes a broad understanding of accessibility and focuses on how barriers to learning are socially constructed in educational contexts (Booth and Ainscow 2003, p. 10). Wocken (2012) uses the metaphor of a pedagogical house, which requires a balance of double adaptation between the learning needs of children and both the pedagogical offers from the environment (didactic adaptation) and the competences of the teachers (professional adaptation) (Wocken 2012, p. 113).

This double adaption is especially relevant in Germany considering that about 40% of students in special schools were diagnosed in 2016 with a special need in learning, making up the largest group requiring special educational support (Ellger-Rüttgardt 2016). 'Special needs' or 'supportive education' and 'learning disabilities' or 'specific learning differences/difficulties' all relate to learners who experience some form of barrier to learning in regular schooling contexts (Heimlich 2016). This paper works with the concept of 'specific learning differences' to highlight the social-cultural aspect of learning where 'deviant behavior' and 'disorders' in the classroom mirror institutionally accepted standards for what is considered the norm or norm-deviant (Gerlach 2015, p. 93).

In the school context, specific learning differences refer to differences in students' responses to learning requirements in a way that learning cannot be tackled without specific pedagogical support within the existing school structure (Heimlich 2016). General areas of difference in learning an additional language are linked to memorization, anxiety, and lexical and grammar confusion (Difino and Lombardino 2004) and should not be understood as having a fixed cut-off point of diagnosis but rather as fluid and varied on a continuum of mild to severe (Gerlach 2019; Kormos 2017; Nijakowska 2010). According to Gerlach (2019), foreign language learners with reading and writing differences display reduced attention due to limited capacity of working memory, especially due to weakness in phonological processing. Reduced attention also negatively influences the use of metacognitive strategies such as planning and the structured processing of tasks (Gerlach 2019, p. 27).

Some SpLDs, such as dyslexia, are directly linked to the process of learning an additional language and include spelling and grammar issues, such as phonetic and inconsistent spelling, the misuse of homophones, transposing letters, the misuse or omission of punctuation, inappropriate grammar constructions and use of tense, or the use of an inappropriate

narrative mode (Westwood 2004). Learners experiencing an SpLD might fail to identify the main points of a text, misunderstand the stated question, or digress from the topic in their answers, their output might lack clarity and be repetitive, and they are often demotivated as a consequence of these difficulties (Kałdonek-Crnjaković 2018; Westwood 2004).

Additionally, general language learning differences can result solely from exogenous factors, such as instructor or curriculum (Difino and Lombardino 2004, p. 393). Especially teachers' in-class behavior, i.e., their teaching methods and attitudes towards SpLDs, play a significant role in shaping learners' motivation and anxiety (Kormos 2017, p. 41). Additionally, an investigation by Markova et al. (2015, p. 2) showed that:

> *Across many countries, students with immigrant backgrounds are disproportionally identified as having special educational needs [ ... ]. Evidence has shown that students from certain ethnic minorities and of lower socioeconomic status are over-represented in special education programs (Gabel et al. 2009) and under-represented in programs for talented students.*

It is, therefore, conceivable that manifestations of specific learning differences in the English language classroom remain unaddressed if the teaching approaches fail to create accessible learning conditions for all students (Cohen 2011, p. 272). Teacher education should, therefore, comprise training for adaptive forms of learning support, as well as have teachers reflect on their roles and actions in reproducing socially constructed barriers to learning (Heimlich 2016).

Adaptive teaching practice can be based on the differentiation of learning objectives, the level of difficulty of classroom tasks, and the degree of support offered (Kormos and Smith 2012, p. 12), as well as on varying teaching methodologies and materials (Daloiso 2017). Pre-service teachers training for inclusive schools should become aware of teaching methods that have been empirically tested and have shown effectiveness in teaching learners with SpLDs. Sparks et al. (1992), for instance, applied multisensory structured language (MSL) instruction for teaching additional languages to learners with SpLD, especially dyslexia. The approach consists of clearly structuring activities with frequent revisions, simultaneously emphasizing writing and pronunciation for the explicit teaching of phonology (Sparks et al. 1992). The results showed that learners improved their oral and written language abilities in both their first and the foreign language (Sparks et al. 1992). Later, a study by Nijakowska (2008), for instance, showed that dyslexic learners' word reading and spelling outperformed the control group after an intervention with the MSL approach (Nijakowska 2008).

Further, concrete, didactic-methodic approaches that have been shown to increase English language learning accessibility to students with SpLDs include creating opportunities for repetition, explicitly teaching phonological and orthographic information and working with mind maps, diagrams, bullet points, pictures, and models (Kormos and Smith 2012, pp. 112–13). Technology, such as computers, spellcheckers, audiobooks, visual presentation software, voice recorders, the internet, specific websites, speech-to-text software, listening aids with a microphone and headset, and voice output systems, among others, can offer essential support in the classroom (Nijakowska 2010, p. 148).

In short, inclusive teachers should not assume that there is a "best method" to teach all learners but have the methodic repertoire to respond to the needs of the learners even if that means relying on structured and gradual approaches to continuous (and, when necessary, repetitive) practice and explicit language instruction (Daloiso 2017; Gerlach 2015).

*1.2. Perceptions of Effective English Language Teachers*

Studies aiming at identifying good teachers, in general, have recognized three levels of teaching quality that directly impact learning processes: (a) instructional, which refers to learners' cognitive activation and engagement; (b) organizational, encompassing classroom management; and (c) emotional–interactional, which includes individual learning support, a positive learning environment, and constructive feedback on errors (Holzberger et al. 2019, p. 802). Lipowsky (2006, pp. 52–53) emphasizes, in addition, that professional experience

and beliefs shape the quality of teaching. He argues that teachers have a considerable influence on the learning development of students through the level of competence that guides teaching actions for cognitive and metacognitive activation and the right balance of direct instruction and cooperative learning (Lipowsky 2006, p. 64). This is in line with Borg's (2003, p. 81) review of studies on teacher cognition, which views teachers as "active, thinking decision-makers who make instructional choices by drawing on complex, practically-oriented, personalized, and context-sensitive networks of knowledge, thoughts, and beliefs". Narrowing the focus down to the ideal English language teacher, research has generally identified the following characteristics: (1) having knowledge and command of the target language; (2) having the ability to organize, explain, clarify, and create interest and motivation; (3) refrain from displaying any form of favoritism nor prejudice; and (4) being available to students (Brosh 1996). Also, Bell (2005) defined effective language teaching as "clear and enthusiastic teaching that provides learners with the grammatical (syntactical and morphological), lexical, phonological, pragmatic, and sociocultural knowledge and interactive practice they need to communicate successfully in the target language" (Bell 2005, p. 260).

In a questionnaire study on the beliefs and attitudes of effective foreign language teachers, Bell (2005) identified a trend in professional consensus related to communicative approaches to language teaching (CLT). CLT approaches were perceived as generally having a positive effect on learning, while error correction or focus on the grammatical form were seen as contributing less to the learning process (Bell 2005, p. 266). Similarly, Brown (2009) examined how teachers and students perceive good foreign language education. Interestingly, teachers agreed that communicative approaches have a positive impact on foreign language learning and that the teaching of grammar should not be done in isolation but within real-world contexts, while students expressed a preference for formal grammar instruction and showed less agreement about group work (Brown 2009, p. 53). Given the disparities between teachers' and students' perceptions, Brown (2009) concluded that teachers need to explicitly discuss learning preferences with their students (Brown 2009, p. 56).

Harris and Duibhir (2011) argued that, while CLT inspired a greater focus on "learner autonomy, the social nature of learning, curricular integration, focus on meaning, diversity, thinking skills, alternative assessment and teachers as co-learners" (Harris and Duibhir 2011, p. 61), it also rejected isolated practice through drills (Harris and Duibhir 2011, p. 69). However, isolated language practice could promote language learning through raising consciousness of grammatical constructions (Ebsworth and Schweers 1997, p. 242). In short, CLT requires learners to infer meaning from context without much focus on the explicit teaching of phonetics and phonology or grammatical rules. For learners experiencing SpLDs, this could mean having a lower chance of success (Nijakowska 2010, p. 127). On the one hand, CLT offers a "comprehensive view of language not restricted to grammatical correctness", which could be favorable for some learners with dyslexia, for instance (Daloiso 2017, p. 70). The stress on communication can still act as a trigger for language anxiety for learners who need explicit language instruction or more time to process information (Daloiso 2017).

Adapting teaching approaches to accommodate a wide variety of learners in the English language classroom might, therefore, require teachers to distance themselves from assumptions regarding the benefits of CLT for English language learning. This does not imply a rejection of the communicative approach, but rather that its implementation should be adjusted to the needs of the learners in a given teaching situation. Nonetheless, the dominance of communicative approaches seems to evoke the conviction among participants that it is a universally effective teaching approach, which carries the danger of making the implementation of inclusion measures in the classroom more difficult (Pfingsthorn 2021, in press).

### 1.3. Differentiation as Adaptivity Competence

The teacher is a central, contextual factor in students' learning processes as a person who can, willingly or unwillingly, directly impact the learning process of their students (Hattie 2009; Pfrang and Viehweger 2015, p. 295). While the individual cognitive factors of the learners or the influence of their family, peers, and school structural factors, which all interact in shaping learning processes, are robust and difficult to change, teachers can adapt their teaching to the given classroom situation (Hattie 2009). This means that structural school changes resulting from educational policies for inclusive education need to be accompanied by changes in the curriculum as well as in teacher education programs that train teachers to incorporate inclusive principles into their teaching practice (Zierer 2015, pp. 23–25). There is evidence to suggest that teacher education programs are necessary for pre-service teachers' professional development, as they stimulate the development of reflection on skills, intuition, and knowledge (e.g., Weinstein 1990). Therefore, the application of inclusive principles into teaching practice requires an understanding of teachers' competence development in terms of their adaptivity to heterogeneous classroom contexts (Beck et al. 2008) and reflexivity to deal with differentiation antinomies that arise from the tension between treating all students equally and the need to differentiate between the group of students and individual students (Helsper 1996).

In the literature on inclusive pedagogies, differentiation has long been identified as a didactic principle (Götz et al. 2015, p. 34) and as a central professional competence: "Differentiation competence is an integrated system of abilities and skills, stocks of thought and knowledge, as well as values and attitudes of an educator that make him/her capable of action to offer instruction adapted to individual learning prerequisites and paths" (Drinhaus and Werner 2015, p. 108). Internal differentiation, as opposed to external forms of schooling differentiation, such as class size and grade level, is the application of individual or selected group support within a learning group. Internal differentiation can include differentiation of goals, curriculum content, teaching formats, and materials. It can either be applied through a top-down and teacher-centered approach or through bottom-up methods, where the students themselves choose their materials and activities (Giesler et al. 2016, pp. 64–65).

In line with the reformist pedagogical ideas of the 1970s, bottom-up differentiation refers to the ways in which teachers prepare the learning environment and support learners if necessary—the teacher is placed in the background of learning and takes the position of an "advisor" to learning. English language education is here centered on communicative tasks and includes open and cooperative learning, such as weekly plan work, free working stations, and project work, placing a strong emphasis on learners' autonomy (Trautmann 2011, pp. 7–9). However, the application of bottom-up differentiation approaches, especially individualization, needs to be conducted in a way that does not lead to the isolation and labeling of individual learners (Wocken 2012, p. 119) as it carries the danger of sustaining exclusionary practices that focus on learners attempting to achieve their "optimal self" with the stated goal of fulfilling normative performance ideals (Idel and Rabenstein 2016, p. 16).

Differentiation competence, as a sub-category of teachers' adaptivity competence, requires not just subject knowledge but also pedagogical diagnostic competences to accurately assess individual learning prerequisites and outcomes to offer appropriate teaching formats (Beck et al. 2008, p. 41). According to profession theory, competence development takes place in an interplay with pedagogically relevant beliefs and teachers' actions (Baumert and Kunter 2006, p. 497). In this sense, teachers' beliefs are intertwined with and support teachers' professional knowledge acquisition (Baumert and Kunter 2006, p. 496). Following this rationale and given the discussion presented above, this paper is interested in: (a) pre-service teachers' perceptions of what good English language education encompasses as a reflection of their professional beliefs; and (b) the extent to which pre-service teachers view various differentiation approaches as positive for the English language learning process. The results should shed light on the knowledge areas that need attention in the first phase of English language teacher professionalization.

## 2. Methods

Insights into participants' cognition of good teaching practice and differentiation approaches were gathered through a questionnaire consisting of two open and forty-eight closed questions. The two open questions of the questionnaire focused on the general conceptualization of "good" and "bad teachers". The rationale behind this choice was the intention to examine what characteristics or aspects of "good" and "bad" teachers are salient in the minds of the participants, without priming them with pre-existing categories. The closed items in the questionnaire examined the extent to which pre-service English language teachers perceived various approaches to differentiated instruction. This included consideration of those that are counter-intuitive to the basic assumptions of the CLT approach to language teaching dominant in the educational context of the participant as potentially having a positive effect on learning. The logic behind this choice of method was to include a wide spectrum of possible methodical classroom choices known and discussed in the literature, instead of relying on the methods salient or accessible to the participants' memory at the time of the study.

The questionnaire ran on an online platform and was filled out voluntarily by forty university students (N = 40) pursuing their BA and M.Ed. degrees in English, which comprise the first qualification phase in the state-certified teacher education program for different school levels in the German states of Bremen and Lower Saxony. The questionnaire was anonymous, and the participants were informed that they could abort their participation without any consequence at any point. Demographic information about gender, majors, levels being studied, and future school types was collected to gain an insight into the population sample of future English language teachers for this study.

The results showed that thirty-five participants were female and five male; no one identified as non-binary. Half of the participants were studying towards a BA degree and the other half towards a M.Ed. degree, all majoring in English and a second or third major. German was the most cited second subject of study, followed by inclusive pedagogy and religion. Most of the participants were studying towards becoming teachers in secondary school forms (Table 1).

**Table 1.** Participants' future school type in total numbers (N = 40).

| School Type | Master | Bachelor | Total |
|---|---|---|---|
| High school | 15 | 12 | 27 |
| Primary school | 2 | 6 | 8 |
| Vocational school | 2 | 2 | 4 |
| Lower schools [1] | 1 | | 1 |
| Total | 20 | 20 | 40 |

[1] Lower schools refer to the German *Haupschule* and *Realschule*, which are school forms that lead to vocational training and do not qualify for tertiary education.

The demographic information collected showed that about a third of the participants were studying to become vocational, lower school, or primary education teachers. The remaining participants were studying to become high school teachers.

The forty-eight closed items comprised approaches to differentiated instruction in the four competences (reading, listening, speaking, and writing) and grammar, as well as general forms of pedagogical and didactic differentiation practice, and presented the participants with the answer options as a 5-point Likert scale of agreement. The items were adapted from the suggestions for differentiation in English language teaching by an Oxford Experts' (Daloiso et al. 2018) publication and validated for correctness and comprehension by two peer researchers. The questionnaire also included some validated items from Bell's (2005) study that measure the behaviors and attitudes of effective English language teachers, as well as some from Kojima's (2017) paper on good English language teacher characteristics. The statistical analysis consisted of descriptive statistics including frequencies, percentages, means, and standard deviations. The categories were all above

the minimum threshold of a Cronbach's alpha of 0.60, displaying internal consistency (Table 2).

**Table 2.** The summed variables and Cronbach's alpha.

| Topic and Number of Variables | Cronbach's Alpha |
| --- | --- |
| Pedagogical differentiation (9 items) | 0.77 |
| Subject-related differentiation (7 items) | 0.66 |
| Differentiation for speaking (6 items) | 0.73 |
| Differentiation for reading (7 items) | 0.66 |
| Differentiation for writing (7 items) | 0.83 |
| Differentiation for listening (7 items) | 0.62 |
| Differentiation for grammar (5 items) | 0.81 |

The qualitative part of the questionnaire asked participants to state if they think that it is possible to talk about "good" and "bad" foreign language teachers and, if so, how these two categories could be defined. The results were analyzed through content analysis based on Mayring's (2003) methodology and followed an inductive systematic analysis of the material. The categories were formed following the seven analytical steps of (1) determination of the units of analysis; (2) paraphrasing the text passages that are important for the content; (3) determining the level of abstraction; (4) reduction by selection, deletion of paraphrases with the same meaning; (5) reduction through bundling; (6) compilation of the new statements as a category system; and (7) re-examination of the summarizing category system using the source material (Kuckartz 2010). The content analysis revealed a total of six categories, three for "good teachers" and three for "bad teachers", which are discussed in the results section below.

## 3. Results and Discussion

### 3.1. Conceptualizations of "Good" and "Bad" English Language Teachers

From the total of 40 participants who filled out the questionnaire, 39 answered the open question on whether they think there are "good" and "bad" foreign language teachers and, if so, how they could be described. Only two participants answered that there are no "bad" FL teachers, and one participant answered that there are no "good" FL teachers. The questions and the answers were given in German, which is the official, local language in which their teacher education program is conducted. Content analysis of the remaining answers revealed conceptualizations of "good" foreign language teachers according to the identified categories below:

> *"Good" FL teachers . . .*
> Category 1: teach inclusively;
> Category 2: are instructionally competent;
> Category 3: motivate.

The first category refers to teaching inclusively. An inclusive teacher, according to the German Society of Foreign Language Research (Gerlach et al. 2021), needs to have knowledge related to the different dimensions of heterogeneity and individual learning requirements and to reflect on their own teaching actions, leading to the adaption of teaching practice to reduce learning barriers (such as materials and didactic methods). Also, teachers should prepare well-structured lessons and offer support to all learners. For inclusive education to properly function, inclusive teachers need to receive professional support from and cooperate with special educators and other professionals (Gerlach et al. 2021). Taking these requirements into consideration, we identified three main aspects in the category "Good FL teachers are inclusive": didactic/methodic knowledge, self-reflection combined with adaptivity of teaching practice, and communication in teams. This can be exemplified by the answers below[1]:

P28: "Good foreign language teachers pay attention to the heterogeneity of their students, strive for a communicative and open teaching culture."

P32: "Reflective, self-aware, communicative, inclusive attitude, knowledge of learning processes, tolerant of ambiguity, self-concept as a facilitator and architect of learning arrangements rather than an instructor."

P30: "Can flexibly adapt themselves and their language to the situation and the learners are willing to express examples differently, communicative cooperative approach, allow learners to make mistakes."

The statement that inclusive teachers "pay attention to the heterogeneity of their students" displays awareness that a classroom is never homogenous and that differences in competences need to be taken into consideration. The participants went on to mention the need for communication or establishing a communicative culture, which highlights the importance of discussion and exchange—possibly with students, potentially also with colleagues. If we assume that "communication" could refer to the interaction in the FL classroom itself, participant 32 seems to have awareness of their role as a facilitator and architect of these processes.

The critical-reflexive posture of the teacher is in line with adaptivity competence and professional identity development perspectives. The participant showed awareness of professionalization discourses for teachers, such as the need to have knowledge of learning processes and being self-aware (cf. Gerlach and Leupold 2019). Still, the participant did not clarify what an "inclusive attitude" entails.

The second category refers to "good" teachers as being "instructionally competent". This is understood here in line with Holzberger et al.'s (2019) findings that good teaching at an instructional level entails being able to activate learners cognitively, being organized and managing the classroom, and giving constructive responses to errors (Holzberger et al. 2019, p. 802). The statements below reflect these findings:

P4: "A good foreign language teacher should have good subject knowledge, be able to explain well, be fair, organized, student-centered, motivating, and available. Also, be in control of the class, be able to activate and use different methods."

P17: "Yes, good foreign language teachers respond to the needs and interests of students, make the lessons varied, and emphasize the teaching of important basics (vocabulary, grammar). They create many opportunities to speak and a positive culture of mistakes."

Both statements refer to teachers responding to the needs and interests of the students, as well as the application of different methods and fair treatment of errors. Participant 17 specifically mentioned the teaching of vocabulary and grammar as "important basics" and was the only participant to make reference to the teaching of these skills, which contradicts, for instance, Bell's (2005) identified effective FL teacher as one who provides learners with grammatical, lexical, phonological, and pragmatic forms of language knowledge and skills (Bell 2005, p. 260).

Additionally, some participants recognized the role of the teacher in making the subject matter accessible, as in the example from participant 1 below. Creating accessibility in learning and recognition of learners' variability and the necessity for teachers to be adaptive to this variability reflects some of the inclusive principles behind the Universal Design for Learning (cf. Meyer et al. 2014):

P1: "Because these are the teachers who have the tools to make language accessible and interesting. No student wants the tenth theory-heavy subject, in which only rules are crammed. Teachers who design language lessons in such a way that they meet the learners where they are oriented to their world and current topics, and also include new methods and concepts, such as task-based or supported learning. Nobody wants to read 20 old novels, but maybe speeches are more interesting and also suitable for learning linguistic means and rhetoric."

Also, the statement could be interpreted as a reference to scaffolding according to Vygotsky's Zone of Proximal Development (Heimlich 2016), in which the participant defined good teachers as those who "meet learners where they are". The participant went on to mention task-based learning, which requires a higher level of autonomy from language learners, but they also referred to the inclusive notion of supported learning.

The third category is about teachers being motivated as well as able to motivate their learners. Motivation is a key issue in teaching inclusively given that learners with an SpLD often suffer from anxiety or develop a negative attitude towards the target language, as mentioned in the discussion above (cf. Kormos 2017). The participants' references to motivation focused on fostering the affective aspect of FL learning ("arouse enthusiasm", "motivate"), as the following quotes depict:

> P13: "Good foreign language teachers design teaching that is varied and up-to-date and should be able to inspire learners to learn new things and to arouse enthusiasm for the foreign language."
> P7: "Good foreign language teachers are open and motivate their students to learn the language."

To sum up, the participants' answers displayed an underlying inclusive orientation throughout by acknowledging the need for self-reflection that leads to adaptive teaching actions and oriented actions, as well as increased accessibility in the classroom. The participants' perceptions of "good" FL teachers reflect a shared knowledge of current discourses on best practices and are in line with the results of previous research on perceptions of good teachers, especially concerning classroom instruction, classroom management, and general didactic knowledge. The teacher as a central factor of learners' extrinsic motivation was often mentioned by the participants, and the category co-occurred with different categories, such as that concerning instructional competence, displaying awareness of the relationship between affective and cognitive processes in learning a FL. Nonetheless, too little attention was given to language forms and the teaching of specific language skills.

The categories identified that describe "bad" foreign language teachers were consistent and, thus, mirrored their characterizations of "good" teachers, whereby a focus on grammar became visible as having a negative effect on FL language learning:

> *Bad Fl teachers . . .*
> Category 1: lack didactic-methodic competence;
> Category 2: create anxiety;
> Category 3: focus mostly on grammar.

Teachers who lack didactic-methodic competence are perceived as being "bad" teachers. Methodic-didactic knowledge and competences are understood here as teachers' capacity to assess the advantages and disadvantages of possible methods and to know when to apply them. This is key to the development of adaptivity competence (Gerlach and Leupold 2019). This category also aligns with the standards of teacher education put forward by the Ministry of Education (2008), which sees, among other things, the necessity of competence development in the areas of language knowledge and didactic design (KMK 2008):

- Teachers should have in-depth knowledge of the foreign language. They should maintain and constantly update their foreign language and intercultural competence;
- Teachers should know the possibilities of designing teaching and learning arrangements, especially considering heterogeneous learning prerequisites.

The participants mentioned failure to properly structure a lesson or differentiate and being incompetent or lacking interest in the FL, as well as not incorporating new or varied methods into their teaching, as concrete examples. These statements especially exemplify many of these expected competences that teachers today should develop:

> P1: "In my opinion, these are the teachers who live in the past with their concept of foreign language teaching. The students are so often in contact with English-language content on social media in their free time that they often know much

more about slang or trends than teachers do. Teachers who do not address this but insist on the standard program are bad."
P20: "When old patterns are persisted in and attempts are made to adapt learners to these patterns instead of vice versa."
P22: "No consideration of didactic concepts (no step-by-step explanation, but tendency to self-study) and starting from homogeneity."

The participants expressed the conviction that adhering to a standard teaching program is not desirable. They rejected the idea of a homogenous classroom, which implies a certain expectation that good teachers need to be open to methodological ideas that exceed the established traditions. Participant 1 made references to different aspects of the standards for teacher education, including the need for teachers to constantly update their FL language competence. The participants seemed to have a broad understanding of what is expected of them by the educational policy in terms of didactic-methodic knowledge, as the statements displayed awareness of central didactic competences.

The second category relates to participants' classifications of "bad" FL teachers as those who make the learners feel anxious. The category "creating anxiety" refers here to the anxiety generated by the teacher, referred to as "state anxiety", which occurs when learners feel threatened by particular tasks and situations that generate feelings of stress and tension (Nijakowska 2010). Participants related that "bad" FL teachers create a "tense learning atmosphere", "scare students", or are "frightening":

P27: "A bad foreign language teacher creates a tense learning atmosphere, is unfriendly."
P39: "Teachers who scare students."
P11: "Frightening, paying too much attention to grammar."

These statements show that what generates state anxiety in students can be different for each participant. While, for P27, anxiety was related to the teacher being unfriendly to the students and, thus, promoting a negative learning atmosphere, the others linked anxiety to the teacher's too strong attention to grammar. Participant 39 did not clarify what is meant by teachers who scare students, so this could be related to any teacher action that results in stress.

The last category identified "bad" teachers as those that "focus mostly on grammar". A strong grammar focus was perceived by the participants as the correction of errors and decontextualized grammar practice. For instance, participants stated that "bad" teachers are:

P13: "Only concerned with correctness in grammar and pronunciation. Ingrained teaching according to 'Formula X'."
P24: "Bad teachers place more emphasis on pure grammar. Bad teachers condemn students who may not find this so easy."
P36: "Teachers who only pay attention to grammar and writing, but do not practice free speaking."

Grammar and correctness were rejected by the participants as they seemed to prefer more open, communicative teaching formats. In this case, the participants did not consider the specific needs in inclusive settings, in which paying attention to grammar or pronunciation might benefit learners who cannot inductively learn these skills.

The results show that there is consensus in viewing good teachers as being didactic-methodically competent, as well as being inclusive in terms of considering learners' interests and generating an accepting and open learning environment. A focus on grammar and form was generally seen as negative, which potentially leaves learners who might need the explicit teaching of form at a disadvantage.

### 3.2. Pre-Service Teachers' Perceptions of Differentiation Approaches

The quantitative part of the questionnaire included 48 items that described concrete practices of differentiation, as well as pedagogical and subject-related teacher competences, in the FL classroom. Participants were asked to decide on a Likert-type scale from 1 (strongly agree) to 5 (strongly disagree) how much each item contributes to effective foreign language teaching. A strong, positive agreement was shown for all items in the pedagogical differentiation categories (Table 3). Items that met with the highest degree of agreement included the notions that effective FL teachers are sensitive to different cultural traditions of learners, recognize that not everyone in the class will produce the same amount of work, spend time with learners who need more encouragement or clarification, value the different skills of the learners, and exchange ideas with other teachers:

**Table 3.** General pedagogical competence.

| The Effective Foreign Language Teacher … | Response Rate % | Mean on Scale of 1–5 | Std. Dev. | Agreement % | Disagreement % | Uncertainty % |
|---|---|---|---|---|---|---|
| is sensitive to the different cultural traditions of the learners. | 97.5 | 1.41 | 0.59 | 94.87 | 0 | 5.13 |
| values the opinions of the learners. | 100 | 1.28 | 0.68 | 92.5 | 2.5 | 5 |
| values the different skills of the learners. | 100 | 1.10 | 0.44 | 95 | 0 | 5 |
| exchanges ideas with other teachers. | 100 | 1.25 | 0.63 | 95 | 2.5 | 2.5 |
| cooperates with other teachers. | 100 | 1.30 | 0.69 | 92.5 | 2.5 | 5 |
| praises the learners for their work. | 100 | 1.30 | 0.76 | 87.5 | 2.5 | 10 |
| recognizes that not everyone in the class will produce the same amount of work. | 100 | 1.28 | 0.51 | 97.5 | 0 | 2.5 |
| accepts that the learning process is more important than the finished product. | 100 | 1.38 | 0.63 | 92.5 | 0 | 7.5 |
| spends time with learners who need more encouragement or clarification. | 100 | 1.33 | 0.47 | 100 | 0 | 0 |

The estimate of general didactical differentiation as having a positive effect on learning a foreign language on a Likert scale (1 = strongly agree; 2 = agree; 3 = partially agree; 4 = do not agree; 5 = strongly disagree). Results as a percentage, whereby 1 and 2 = agreement; 3 = uncertainty; 4 and 5 = disagreement.

A strong, positive agreement was also shown for most items in the general, subject-related differentiation categories (Table 4), except for "arranging group work so that there is a balance of abilities" and "letting learners choose the topics to be covered", with which the participants agreed slightly less strongly. In addition, the participants agreed with the idea that the effectiveness of FL teachers can be associated with their ability to identify different types of (possible) barrier to learning and with the provision of a variety of tasks at different levels in the classroom.

**Table 4.** General subject related differentiation.

| The Effective FL Teacher . . . | Response Rate % | Mean on Scale of 1–5 | Std. Dev. | Agreement % | Disagreement % | Uncertainty % |
|---|---|---|---|---|---|---|
| gives tasks that allow for easy successes at the beginning to motivate learners. | 100 | 1.50 | 0.68 | 90 | 0 | 10 |
| is able to identify the characteristics of different types of (possible) barriers to learning. | 100 | 1.23 | 0.42 | 100 | 0 | 0 |
| is adept at providing a variety of tasks at different levels in the classroom. | 100 | 1.23 | 0.48 | 97.5 | 0 | 2.5 |
| arranges group work so that there is a balance of abilities. | 100 | 1.63 | 0.84 | 87.5 | 5 | 7.5 |
| continuously evaluates their teaching. | 100 | 1.18 | 0.38 | 100 | 0 | 0 |
| adapts their teaching approach to meet the needs of the learners. | 100 | 1.23 | 0.53 | 95 | 0 | 5 |
| allows learners to choose the topics to be covered. | 97.5 | 1.69 | 0.73 | 84.62 | 0 | 15.38 |

The estimate of general, subject-related differentiation as having a positive effect on learning a foreign language on a Likert scale (1 = strongly agree; 2 = agree; 3 = partially agree; 4 = do not agree; 5 = strongly disagree). Results as a percentage, whereby 1 and 2 = agreement; 3 = uncertainty; 4 and 5 = disagreement.

The Cronbach's alpha revealed sufficient internal consistency (0.62) among the following items for them to be perceived as aspects of the construct "differentiation approaches to listening" (Table 5). Participants agreed with almost all items of the listening differentiation category with an agreement of at least 67.57% on all items except "lets learners know in advance what questions will be asked so they can prepare" which had 58.97% agreement.

**Table 5.** Differentiation approaches for listening.

| The Effective FL Teacher . . . | Response Rate % | Mean on Scale of 1–5 | Std. Dev. | Agreement % | Disagreement % | Uncertainty % |
|---|---|---|---|---|---|---|
| allows additional time for learners to answer questions. | 95 | 1.26 | 0.50 | 97.37 | 0 | 2.63 |
| repeats questions several times slowly. | 97.5 | 2.05 | 0.92 | 71.79 | 2.56 | 25.64 |
| rephrases questions using simpler language. | 97.5 | 1.56 | 0.68 | 89.74 | 0 | 10.26 |
| lets learners know in advance what questions will be asked so they can prepare. | 97.5 | 2.28 | 1.12 | 58.97 | 15.38 | 25.64 |
| uses post-it notes to capture ideas. | 92.5 | 2.16 | 0.99 | 67.57 | 8.11 | 24.32 |
| offers instructions on tasks in a visual format, such as bullet points. | 97.5 | 1.87 | 1.03 | 76.92 | 7.69 | 15.38 |
| asks learners to underline the keywords of the rubrics in listening tasks. | 97.5 | 1.87 | 0.83 | 82.05 | 5.13 | 12.82 |

The estimate of differentiation approaches for listening as having a positive effect on learning a foreign language on a Likert scale (1 = strongly agree; 2 = agree; 3 = partially agree; 4 = do not agree; 5 = strongly disagree). Results as a percentage, whereby 1 and 2 = agreement; 3 = uncertainty; 4 and 5 = disagreement.

There was relatively strong agreement that measures that alleviate potential learning difficulties in listening comprehension and following instructions, such as repeating and rephrasing questions in simpler language, offering visual instructions, highlighting structure in terms of keywords, or guiding learners' attention to concrete aspects of the text are indeed welcome in the FL classroom. This is in line with some of the suggestions made in the Oxford ELT Expert Panel's position paper on inclusive education (Daloiso et al. 2018).

Pre-service teachers also agreed with all items related to the differentiation categories for reading (Table 6), and the item with the least agreement (69.23%) was 'lets learners select reading material that matches their competence level'.

**Table 6.** Differentiation approaches for reading.

| The Effective FL Teacher ... | Response Rate % | Mean on Scale of 1–5 | Std. Dev. | Agreement % | Disagreement % | Uncertainty % |
|---|---|---|---|---|---|---|
| lets learners select reading material that matches their competence level. | 97.5 | 2.00 | 0.92 | 69.23 | 5.13 | 25.64 |
| encourages discussion of the topic before reading. | 95 | 1.76 | 0.85 | 78.95 | 2.63 | 18.42 |
| helps learners use all available contextual information from a text. | 95 | 1.58 | 0.68 | 89.47 | 0 | 10.53 |
| highlights cultural elements of a text to make them explicit. | 95 | 1.76 | 0.82 | 81.58 | 2.63 | 15.79 |
| uses technology to support the reading of longer passages. | 95 | 1.92 | 0.91 | 73.68 | 5.26 | 21.05 |
| breaks down texts into smaller parts. | 95 | 1.84 | 0.68 | 84.21 | 0 | 15.79 |
| engages learners in comprehension tasks immediately after reading each part of a text. | 95 | 1.82 | 0.80 | 81.58 | 2.63 | 15.79 |

The estimate of differentiation approaches for reading as having a positive effect on learning a foreign language on a Likert scale (1 = strongly agree; 2 = agree; 3 = partially agree; 4 = do not agree; 5 = strongly disagree). Results as a percentage, whereby 1 and 2 = agreement; 3 = uncertainty; 4 and 5 = disagreement.

The results presented in Table 6 show relative agreement with most of the differentiation techniques that can address various needs of learners. The Oxford ELT Panel (Daloiso et al. 2018, p. 39) suggests that teachers "encourage discussion of the topic before reading; help students to make use of all the contextual information available; highlight cultural elements of the text to make them clear for all" when reading processes are inaccurate or comprehension is incomplete. Breaking down the text into smaller parts and directing learners' attention to comprehension tasks directly after reading have been shown to be promising strategies for learners who need more time to process a text than their peers (Daloiso et al. 2018, p. 40).

However, the practical classroom application of differentiation that contrasts with dominant communicative methodologies showed less general acceptance among the participants involved in this study. This is especially the case for differentiation approaches for speaking (Table 7), where the participants only partially agreed that spending time on wording can have a positive impact on foreign language learning (48.72%), and the vast majority (95%) agreed with the potential of employing project work to help learners achieve communicative goals. Also, 70% agreed that fluency should be prioritized over correctness.

**Table 7.** Differentiation approaches for speaking.

| The Effective FL Teacher . . . | Response Rate % | Mean on Scale of 1–5 | Std. Dev. | Agreement % | Disagreement % | Uncertainty % |
|---|---|---|---|---|---|---|
| fluency is prioritized over correctness. | 100 | 2.18 | 1.22 | 70 | 15 | 15 |
| idiomatic expressions are taught so learners can successfully hold conversations in the target language. | 90 | 1.89 | 0.89 | 72.22 | 2.78 | 25 |
| employs project work to help learners achieve communicative goals. | 100 | 1.55 | 0.60 | 95 | 0 | 5 |
| encourages learners to begin speaking in the target language only when they are ready. | 100 | 2.48 | 1.13 | 60 | 17.5 | 22.5 |
| spends focused time on wording. | 97.5 | 2.31 | 0.92 | 48.72 | 5.13 | 46.15 |
| gives explicit instructions on how to produce the sounds in isolation. | 95 | 2.39 | 1.08 | 63.16 | 15.79 | 21.05 |

The estimate of differentiation approaches for speaking as having a positive effect on learning a foreign language on a Likert scale (1 = strongly agree; 2 = agree; 3 = partially agree; 4 = do not agree; 5 = strongly disagree). Results as a percentage, whereby 1 and 2 = agreement; 3 = uncertainty; 4 and 5 = disagreement.

Interestingly, in the differentiation approaches for the development of written competence presented in Table 8 below, the participants disagreed with the use of technology to support foreign language learning; only 33.33% of participants thought that speech-to-text software has a positive effect on learning, and only 42.86% agreed that using technologies, such as word processing instead of handwriting, can foster learning. Also, they expressed ambiguous views about the use of paper with guiding elements such as margins (37.14%). Nonetheless, teachers generally agreed with helping learners develop planning strategies to first capture and later write down their ideas (89.19%), providing memory cues to help learners remember difficult parts of irregular words (94.74%), and teaching common spelling patterns (84.21%). Slightly fewer agreed to drawing attention to patterns of morphology and syntax, with 62.6% agreement.

**Table 8.** Differentiation approaches for writing.

| The Effective FL Teacher . . . | Response Rate % | Mean on Scale of 1–5 | Std. Dev. | Agreement % | Disagreement % | Uncertainty % |
|---|---|---|---|---|---|---|
| teaches common spelling patterns. | 95 | 1.97 | 0.75 | 84.21 | 5.26 | 10.53 |
| draws attention to patterns of morphology and syntax. | 92.5 | 2.19 | 0.94 | 62.16 | 8.11 | 29.73 |
| provides memory cues to help learners remember difficult parts of irregular words. | 95 | 1.47 | 0.60 | 94.74 | 0 | 5.26 |
| helps learners develop planning strategies to first capture and later write down their ideas. | 92.5 | 1.57 | 0.69 | 89.19 | 0 | 10.81 |
| advocates the use of paper with guiding elements such as margins, spacing, etc. | 87.5 | 2.57 | 1.20 | 45.71 | 17.14 | 37.14 |
| allows the use of technology, e.g., word-processing software instead of handwriting. | 87.5 | 2.75 | 1.08 | 42.86 | 17.14 | 40 |
| allows the use of speech-to-text software. | 90 | 3.08 | 1.23 | 33.33 | 38.89 | 27.78 |

The estimate of differentiation approaches for writing as having a positive effect on learning a foreign language on a Likert scale (1 = strongly agree; 2 = agree; 3 = partially agree; 4 = do not agree; 5 = strongly disagree). Results as a percentage, whereby 1 and 2 = agreement; 3 = uncertainty; 4 and 5 = disagreement.

These results point towards a lack of awareness that, in some cases of specific learning differences, learners might benefit from the use of technological support (Wood et al. 2018; Nijakowska 2010).

In relation to the differentiation approaches for grammar (Table 9), the results are in line with the answers to the open questions of the questionnaire. Here, too, the participants did not share the belief that grammar teaching has a positive effect on foreign language learning. Only 53.85% agreed with creating lesson plans that emphasize grammatical aspects, and 38.46% agreed with teaching grammar deductively. On the other hand, 87.18% agreed that teachers should give activities that focus learners' attention on specific grammar features. In terms of using the senses to make grammar learning more accessible, 79.49% agreed that it is good to create mind maps for the visualization of language information. On the other hand, only 40% seemed to think that singing example sentences with the grammatical structures to be learned helps learners learn grammar.

**Table 9.** Differentiation approaches for grammar.

| The Effective FL Teacher . . . | Response Rate % | Mean on Scale of 1–5 | Std. Dev. | Agreement % | Disagreement % | Uncertainty % |
|---|---|---|---|---|---|---|
| creates mind maps for visualizing language information. | 97.5 | 1.82 | 0.97 | 79.49 | 5.13 | 15.38 |
| sings example sentences that contain the grammatical structure to be learned. | 87.5 | 2.94 | 1.16 | 40 | 31.43 | 28.57 |
| gives activities that focus learners' attention on specific grammar features. | 97.5 | 1.69 | 0.77 | 87.18 | 2.56 | 10.26 |
| creates lesson plans that emphasize grammatical aspects of the target language. | 97.5 | 2.36 | 0.99 | 53.85 | 12.82 | 33.33 |
| teaches grammar deductively (i.e., grammatical rules before examples). | 97.5 | 2.79 | 1.03 | 38.46 | 23.08 | 38.46 |

The estimate of differentiation approaches for grammar as having a positive effect on learning a foreign language on a Likert scale (1 = strongly agree; 2 = agree; 3 = partially agree; 4 = do not agree; 5 = strongly disagree). Results as a percentage, whereby 1 and 2 = agreement; 3 = uncertainty; 4 and 5 = disagreement.

To sum up, even though there was a strong agreement with the items on general pedagogical competence, the questionnaire seemed to generally point towards a limited knowledge of the specific techniques that could be beneficial in supporting FL education to learners with SpLDs. In general, the participants seemed to be aware of broad, inclusive principles but expressed skepticism towards specific support, such as working with paper with guiding elements or the use of supportive technology. Also, their reluctance towards teaching grammar deductively could be interpreted as a reflection of the assumptions of CLT in their perceptions of good FL teaching practice.

## 4. Conclusions and Recommendations

The results of the study point to the conclusion that the participants generally have an inclusive orientation and share a positive stance towards several inclusive principles, such as recognition and acceptance of diversity (Gerlach et al. 2021). There was a consensus among the participants that good FL teachers need to adapt to learner needs and interests in a motivating way. The participants also accepted the notion that spending time with learners who need more encouragement or clarification can have a positive effect on the learning process. They also accepted the assumptions that not all students will produce the same amount of work and that different skills of learners should be valued. In addition, the participants viewed the ability to self-reflect and be open to criticism as important. They

also seemed to believe that good FL teachers need to be equipped with a diversified, subject-related methodic repertoire and know how to deal with language errors in a productive manner.

The participants' conceptualization of "good" and "bad" foreign language teachers reflects traits identified in previous studies, such as the importance of teachers' command of the target language, their ability to create interest and motivation, and their availability to students (Brosh 1996). They seem to strongly reflect Holzberger et al.'s (2019) emotional–interactional aspect of quality teaching in terms of creating a positive learning environment and focusing on constructive responses to error correction (Holzberger et al. 2019, p. 802). This shows that their existing beliefs about differentiation, which form the basis of their adaptivity, relate mainly to more traditional, mainstream teacher qualification and foreign language didactic discourses, such as those focused on communicative tasks.

The data revealed indications that communicative language teaching acts as the guiding principle that influences how the participants view various methodological classroom choices. While, for instance, project work was readily accepted as a valuable approach to foster the achievement of communicative goals, placing fluency over accuracy, a more explicit, deductive focus on grammar and spelling patterns was less favored but not completely rejected.

These findings are likely due to the relatively low awareness of specific learning differences and their possible manifestations, as well as of the possible positive effect of top-down differentiation approaches (e.g., Wocken 2012). The quantitative part of the questionnaire showed that the participants' understanding of differentiation itself was, indeed, framed around ideas of bottom-up differentiation, implying that learners are autonomous and able to decide on their own what is to be learned. This assumes a metacognitive and self-structuring capacity for foreign language learning that many learners experiencing SpLDs may lack (e.g., Heimlich 2016).

There are a few methodological issues that suggest caution in the interpretation of the data. Firstly, as the participation in the study was voluntary, the obtained data set is comprised of the responses of the individuals who were willing to grant this form of indirect access to their cognition related to the concept of "good" and "bad" FL teachers and differentiation techniques. This limits the generalization of the conclusions to those individuals willing to share their thoughts. Secondly, while teacher cognition is an important element influencing actual teaching practice (Borg 2011), teacher beliefs and convictions alone are not sufficient pieces of information to predict actual classroom choices. It is important to emphasize that it may be easier to accept propositions on a conceptual level than to implement them in educational practice. Haug (2016) describes this discrepancy in connection to the implementation of inclusive principles in education as an ideal that is "easy to accept and difficult to be against or even criticize" (Haug 2016, p. 207). The intricacies of the relationship between FL teachers' cognition and their behavior in inclusive classrooms are a research area that still seems to be relatively underexplored.

Thirdly, any form of educational data interpretation, as well as recommendations to be made for teacher education, must first consider the structural limitations that might be in place for its implementation. Since Germany's educational system follows a federalist structure, there is strong variation in the application of inclusive schooling in the different states. Pre-service teachers who receive their education elsewhere are trained to respond to the needs of the local educational systems, which may differ from the one in which this study was set. The state of Bremen, where this study was conducted, has been successively implementing inclusion in all schools, including the establishment of centers for supportive pedagogy (ZuP) in primary and secondary schools. A ZuP offers general special education support, including dyslexia and language learning support. Even though ZuP teachers can offer specific support to learners and teachers, no clear guidelines have been set specifically for teaching foreign languages, and not every school has its own ZuP[2].

Nijakowska (2010) argues that the negative consequences of failing to properly implement regulation into inclusive teaching practice could lead to learners' being exempt from

having to take a FL at school to avoid failing at it in the first place (Nijakowska 2010, p. 146). This is extremely problematic, especially in light of Halliday's (1999, p. 269) assertion that, since all educational learning is mediated through language, either as "a medium of learning" or as "the substance of what is being learned", language is a key competence in all education. Exclusion from language learning can generate wideset negative consequences for the learners, such as missing out on developing valuable FL competences or being denied further educational opportunities at university level. Thus, teachers' adaptivity competence development should be coordinated with legal recommendations targeting the implementation of inclusive FL education specifically.

**Author Contributions:** Conceptualization, A.R. and J.P.; methodology, A.R. and J.P.; validation, A.R. and J.P.; formal analysis, A.R.; writing—original draft preparation, A.R.; writing—review and editing, J.P.; project administration, J.P.; funding acquisition, J.P. All authors have read and agreed to the published version of the manuscript.

**Funding:** This research was supported by the Central Research Funds of the University of Bremen.

**Institutional Review Board Statement:** Not applicable.

**Informed Consent Statement:** Informed consent was obtained from all subjects involved in the study.

**Data Availability Statement:** Data supporting reported results are archived at the University of Bremen.

**Conflicts of Interest:** The authors declare no conflict of interest.

## Notes

[1]   Translated by the authors.
[2]   Senatorin für Kinder und Bildung: https://www.bildung.bremen.de/inklusion-4417 (accessed on 9 June 2022).

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
