# Peer review of "“Good Foreign Language Teachers Pay Attention to Heterogeneity”: Conceptualizations of Differentiation and Effective Teaching Practice in Inclusive EFL Classrooms by German Pre-Service Teachers"

_languages, doi:10.3390/languages7030162_

Round 1

Reviewer 1 Report

The study focuses on an important topic with clear relevance for language teaching and learning.  I find the text generally interesting (esp. Introduction), but I would like to recommend that revisions are made, as outlined below, to clarify and further improve the manuscript.

  1. Introduction (up and until 1.4): Some (more) problematization of existing theory and research (with more recent publications: last paper referenced was published in 2019!). The authors need to include the work of Joanna Nijakowska (https://scholar.google.pl/citations?hl=pl&user=jbVrdjwAAAAJ&view_op=list_works&sortby=pubdate) and more extensively of Judith Kormos (https://www.lancaster.ac.uk/linguistics/about/people/judit-kormos) leading figures in the field of SpLD in English as a second/foreign language.

  1. Methodology: The authors should provide a more clear rationale for the choice of research method, data instrumentation and validation, data collection procedures and append the questionnaire and interview guide in the appendix. Table 1: Cronbach’s alpha should be rounded in two, maximum, decimal values

  1. Results: This is where most of the issues are. First because of the presentation of qualitative results: extracts are listed as if the authors cannot make up their mind as to which ones are illustrative of the comments they make. As a result the authors end up reading numerous extracts and having to make sense/interpret them by themselves. I suggest the authors either shorten the list or they provide enough commentary and reflection before/after the choice of the extracts. Figures are not clear. I suggest they are replaced by tables and titles/legends are reduced to the minimum. Also why do authors keep all these tables in the Appendix? Since they are discussed in the main text, they need to be brought in the text. But only what is needed. The text should include the tables with info that is needed.

  1. Discussion and Conclusion: this is rushed. There are tendencies and interesting findings in the Figures/Tables that have not been discussed. Also the authors need to consider recommendations for research too and reflect on the shortcomings/challenges of their study explicitly

There are some language issues (typos, concord, phraseology, punctuation), which however do not impede comprehension. However, the author tends to write in very long sentences at times which makes it difficult to process.   

Reviewer 2 Report

Thanks for giving me the opportunity to read and comment on this manuscript. Overall I find this to be a well written paper of high scholarly quality which gives an important contribution to the field. However, as special needs education in FL teaching is not my special area of research, I may not be aware of relevant research that should have been referred to.

i have two concerns, both regarding methodology. The questionnaire seems  to have been made availabe online. Does this mean that the informants were those who chose to take part? Or was choice of participants made in another way, and if so, were there individuals who declined to participate? This should be made clearer in the methodology part and reflected on in the discussion. In what ways may this have affected the outcome. I also would like to see some reflections on ethical issues.

Round 2

Reviewer 1 Report

The authors of the manuscript have addressed the feedback comments thoroughly and systematically and submitted an improved version to the satisfaction of the current reviewer. Looking forward to its publication. Thank you

Author Response

Dear Reviewer,

Please find our point by point response in the document attached.

Best regards,

Ana Rovai & Joanna Pfingsthorn
